# Alcohol Use Intensity Decreases in Response to Successful Smoking Cessation Therapy

**DOI:** 10.3390/genes13010002

**Published:** 2021-12-21

**Authors:** Robert Philibert, Kelsey Dawes, Willem Philibert, Allan M. Andersen, Eric A. Hoffman

**Affiliations:** 1Department of Psychiatry, University of Iowa, Iowa City, IA 52242, USA; kelsey-dawes@uiowa.edu (K.D.); willem-philibert@uiowa.edu (W.P.); allan-andersen@uiowa.edu (A.M.A.); 2Behavioral Diagnostics LLC, Coralville, IA 52241, USA; 3Department of Biomedical Engineering, University of Iowa, Iowa City, IA 52242, USA; eric-hoffman@uiowa.edu; 4Department of Radiology, University of Iowa, Iowa City, IA 52242, USA

**Keywords:** smoking, alcohol dependence, DNA methylation, smoking cessation, epigenetics

## Abstract

Smokers frequently drink heavily. However, the effectiveness of smoking cessation therapy for those with comorbid alcohol abuse is unclear, and the content of smoking cessation programs often does not address comorbid alcohol consumption. In order to achieve a better understanding of the relationship between changes in rate of smoking to the change in intensity of alcohol consumption, and the necessity for alcohol-specific programming for dual users, we quantified cigarette and alcohol consumption in 39 subjects undergoing a 3-month contingency management smoking cessation program using recently developed DNA methylation tools. Intake alcohol consumption, as quantified by the Alcohol T Score (ATS), was highly correlated with cg05575921 smoking intensity (adjusted *R*^2^ = 0.49) with 19 of the 39 subjects having ATS scores indicative of Heavy Alcohol Consumption. After 90 days of smoking cessation therapy, ATS values decreased with the change in ATS score being highly correlated with change in cg05575921 smoking intensity (adjusted *R*^2^ = 0.60), regardless of whether or not the subject managed to completely quit smoking. We conclude that alcohol consumption significantly decreases in response to successful smoking cessation. Further studies to determine whether targeted therapy focused on comorbid alcohol use increases the success of smoking cessation in those with dual use should be explored.

## 1. Introduction

For decades, researchers and clinicians have appreciated that smokers often drink and drinkers often smoke [1]. The exact proportion of dual users has varied over the years. In 2014, about 70% of all smokers in the United States reported drinking at least occasionally [2]. The majority of studies that examined dual use used the self-report of both alcohol and cigarette consumption [3,4,5]. However, it is generally appreciated that research subjects underreport alcohol use [6,7]. This underreporting of alcohol consumption and high co-occurrence makes exact attributions of risk for certain health outcomes difficult. For example, both alcohol use and smoking are associated with colon cancer, but the degree of their relative risk is not well established [8,9].

A second problem posed by the frequent comorbidity of smoking with drinking is the impact of alcohol consumption on the success of smoking cessation therapy. Research suggests that dual users are considered to be more refractory to treatment than those who only smoke [10]. However, the exact dose dependency of this second form of substance use on the effectiveness of smoking cessation therapy is unclear. A major contributor to this uncertainty is that while trials of smoking cessation employed biomarkers such as cotinine to verify the quantify of cigarette consumption, to the best of our knowledge, no studies of smoking cessation used biomarkers of alcohol consumption to assess initial alcohol consumption levels or verify changes in alcohol consumption in response to therapy. Instead, almost every study of smoking cessation in dual users has relied on the self-report of alcohol consumption. If this self-report of alcohol use is inaccurate, our understanding of treatment effectiveness may be impacted. Unfortunately, numerous studies have shown that the self-report of alcohol consumption in clinical populations is unreliable [11,12,13].

Conceivably, a more exact understanding of the relationship(s) between smoking and alcohol use could be obtained through the use of biomarkers. However, the current generation of alcohol biomarkers is poorly suited for use in most research studies. The gold standard for assessing alcohol consumption in clinical trials is carbohydrate-deficient transferrin (CDT) [14,15,16]. However, this test is known to have limited sensitivity and requires venipuncture to obtain serum. Alternatively, investigators could employ assessments of phosphatidylethanolamine (PEth), which, in some studies, has outperformed the CDT in detecting recent heavy alcohol consumption (HAC) [16]. However, these PEth assessments also require phlebotomy and special sample handling requirements. As a result, neither of these methods are routinely used to assess alcohol consumption in clinical trials.

The use of DNA methylation to assess alcohol consumption status may be a better alternative for clinical studies of cigarette and alcohol use. Recently, in a blinded study conducted with a well-known insurance testing laboratory, we showed that a metric referred to as Alcohol T-Score (ATS) is more accurate than the CDT in predicting HAC [15,17]. What is more, different from CDT and PEth testing, the ATS can be assessed using DNA prepared from either blood or saliva [18]. Since a recently established biomarker for smoking [19], cg05575921 methylation, which can also be assessed using saliva [20], can be used simultaneously to measure cigarette consumption, this greater accuracy and ease of assessment makes the ATS an attractive metric for helping to decipher the relationship between forms of tobacco use and alcohol consumption. Indeed, in recent work, we used the combination of cg05575921 methylation and the ATS to show that smokers, but not vapers, have a higher rate of HAC, as compared to the general population [21].

The aim of this study is to test whether these prior findings are generalizable to other groups, such as clinical populations. In this paper, we extend those prior cross-sectional findings of alcohol and cigarette use to this examination of smoking- and alcohol-related DNA methylation measures in DNA from 39 subjects who underwent 90 days of incentive-based smoking cessation therapy.

## 2. Materials and Methods

The data presented in this study are derived from 39 subjects who participated in a 90-day incentive-based smoking cessation program at the University of Iowa (National Clinical Trials NCT02682147). All protocols and procedures used in this study were approved by the University of Iowa Institutional Review Board (IRB201706713).

These subjects were recruited through a series of advertisements, distributed to patients and staff of the University of Iowa, which sought regular cigarette smokers over the age of eighteen who were interested quitting. To qualify for the study, subjects had to complete a brief online survey on their smoking habits, report smoking at least 10 cigarettes/day and having a total of five or more pack years of lifetime consumption. Those who qualified were then invited to participate in the smoking cessation protocol.

The study protocol was part of a larger effort to understand the effects of smoking on pulmonary inflammation [22]. The protocol included one intake visit and three consecutive monthly follow-up visits. At each visit, subjects were asked about their current levels of smoking and phlebotomized to provide DNA and serum for the epigenetic and serological analyses. To minimize interference with the medication and of the imaging assessments, subjects were instructed to quit smoking without the aid of smoking cessation pharmacological agents including nicotine replacement therapy. However, each subject was provided smoking cessation treatment, which included brief motivational interviewing support and contingency management. Specifically, each subject was offered USD 400 if they quitted smoking, as evidenced by denying recent cigarette use and having negative serum cotinine levels at 30-, 60- and 90-day visits.

The DNA was prepared from whole blood using cold protein precipitation as previously described [15,20,21,23,24]. Serum cotinine levels were determined by the University of Iowa Diagnostic Laboratories under standard clinical processes. All laboratory procedures were conducted by personnel blinded to treatment outcomes.

DNA methylation assessments of cigarette and alcohol consumption were conducted as previously described [25]. The determination of DNA methylation at cg05575921 and the four loci comprising the Alcohol T Score were conducted using universal droplet digital PCR reagents and equipment from Bio-Rad (Hercules, CA, USA) and propriety primer-probe sets specific for each CpG from Behavioral Diagnostics LLC (Coralville, IA, USA). The Alcohol Signature is a battery of assays that assess methylation status at four loci that are highly responsive to alcohol consumption and are not affected by smoking: cg02583484, cg04987734, cg09935388 and cg04583842 [15]. The ATS metric is an unweighted average of alcohol-induced changes in the T-score (TS) for the four loci [15]. The TS for each CpG site were produced by subtracting the previously established mean of methylation at the locus in abstinent controls from the observed methylation value for each subject at each locus, then dividing the result by the standard deviation of the controls at that locus [15]. Please note that methylation at cg04987734 and cg04583842 increased in response to alcohol consumption, while cg02583484 and cg09935388 demethylated in response to alcohol consumption.

The data were analyzed using the suite of general linear model analytic algorithms embedded in JMP Version 16 (SAS Institute, Cary, SC, USA) and R Studio Version 1.3.959. Comparisons between groups were conducted using *t*-tests for continuous variables and chi-square for categorical variables. Univariate linear regressions were used to assess the relationship between continuous variables.

## 3. Results

The demographic and clinical characteristics of the 39 subjects who completed all four consecutive monthly visits are provided in Table 1. In brief, the subjects averaged 40 years of age, a slight majority were male, and they all were almost exclusively of European ancestry. At intake, the average self-reported cigarette consumption of the subjects over the prior month was 20.9 ± 11.2 cigs/day with a total history of consumption of 28.2 ± 19.8 pack years.

Of the 39 subjects who completed all four visits, 20 successfully quit smoking, as evidenced by the denial of recent cigarette use and having negative serum cotinine (<7 ng/mL) at the 30-, 60- and 90-day post intake visits. There were no significant differences in age, ethnicity, and self-reported cigarette levels between those who successfully quit and those who did not.

Given the focused nature of the online interview and the focus on imaging, the self-report of alcohol consumption was not assessed in the intake interviews. However, the objective ATS scores of the subjects indicated substantial alcohol consumption. Figure 1 shows the distribution of ATS scores in the 39 subjects. The overall average was 3.7 ± 2.9, with 18 of the subjects having a score of 4 or higher, which was highly predictive of current HAC. Figure 2 shows the relationship of ATS score to cg05575921 smoking intensity at study intake. Increasing alcohol consumption, as shown by the ATS score, was highly correlated with increasing smoking intensity (adj *R*^2^ = 0.49, *p* = 3.99 × 10^−7^ and adj *R*^2^ = 0.39, *p* = 1.4 × 10^−5^, respectively).

Table 1 shows the ATS values for the quitting and non-quitting group at study intake and exit. Although there was a slight arithmetic difference, there was no significant different difference of overall group intake ATS scores between those who did not quit (4.22 ± 2.95) and those who succeeded (3.26 ± 2.78) at intake; nor did intake ATS score predict the likelihood of absolute smoking cessation (logistic regression, *p* < 0.28). However, after 90 days of smoking cessation, those who completed all four visits and succeeded in quitting had a significantly lower ATS than those who did not (2.18 ± 2.33 vs. 4.15 ± 3.03, *p* < 0.03).

Figure 3 shows a more granular view of the relationship between changes in smoking behavior and change in alcohol consumption. As indicated by the ATS, even if they were not successful in completely quitting smoking, the vast majority of subjects (29 of 39) decreased their alcohol consumption (delta ATS > 0) over the ninety days of the study. Furthermore, the change in smoking intensity was highly associated with the degree of change in ATS score over the ninety period (*p* < 0.0001, adj *R*^2^ = 0.70). Although the magnitude of change in cg05575921 was greater in those who fully quit smoking (−7.6 ± 5.8% vs. −2.1 ± 5.5; *p* < 0.005), the relationship between change in cg05575921 and change was present and highly significant for both those who managed to fully quit smoking (*p* < 0.0001; adjusted *R*^2^ = 0.62) as well as those who attempted to quit smoking (*p* < 0.0001; adjusted *R*^2^ = 0.70).

## 4. Discussion

Clinicians intuitively appreciate that patients seeking substance use treatment are not a monolithic entity. Nonetheless, deciphering the type and extent of the comorbidities for any given patient is critical for the optimal design of a substance use treatment plan. Innovations that allow clinicians to precisely define the type and extent of comorbid substance use may allow for the better assignment of treatment and reveal other hidden clinical variables that may impede therapeutic efforts.

The current results clearly showed that a cohort of subjects from a Midwest academic center seeking smoking cessation had a high rate of HAC consumption. This may be surprising to some. Nonetheless, smoking and drinking are well known to be frequently comorbid and the finding that the heaviest smokers tend to be the heaviest drinkers is fully supported by the existing literature. Regardless, the strong correlation between cg05575921 indicated smoking intensity and ATS score (*r* = 0.84) is surprising at first glance. However, the overall average ATS score and cg0557921 intensity in this small group of smokers (3.7 and 52%) is very similar to that of a larger group of daily community smokers (*n* = 108, 3.7 and 55%, respectively) from Iowa who were not ascertained for their interest in smoking cessation. Hence, this high rate of comorbid smoking and drinking may be the rule rather than the exception for patients in the Midwest and perhaps throughout the United States.

ATS scores markedly decreased in response to smoking cessation therapy. Although those who completely quit smoking had the greatest magnitude of change, the intensity of HAC decreased as a function of change in smoking intensity, regardless of whether or not a patient managed to completely quit smoking. This finding is critically important for two reasons. First, clinicians are often disappointed when a patient does not absolutely quit smoking. Our study clearly shows that many of those “failures” actually reduce both cigarettes and consumption. Since methylation intensities for smoking and drinking are potent predictors of mortality [26], this suggests that these partial successes truly exist and that the quantitative reduction in harm that these changes in methylation actually represent should instead be celebrated. Indeed, we believe that if the patient could be informed of these changes through a program of Precision Epigenetics, it may be possible to further encourage the patient to make the changes necessary to achieve full and lasting lifetime cessation in smoking.

In addition, these findings show the hidden benefits of smoking cessation. Prior to the global pandemic, excessive alcohol consumption was the third leading preventable cause of death in the United States [27]. However, concurrent with their smoking cessation, these subjects markedly decreased their intensity of alcohol consumption even in the absence of therapeutic programming to address co-morbid alcohol use. Conceivably, if informed of the prospective use of epigenetic information, the skillful employment of targeted therapies to address comorbid alcohol use could further increase the likelihood of total substance use cessation, while not overburdening and possibly alienating those patients who only need smoking cessation.

Although we did not find that HAC influenced the likelihood of achieving smoking cessation in this study, prior studies of larger populations concluded that heavy drinking does influence the likelihood of smoking cessation [4]. Though regrettable, this also presents an opportunity for those interested in refining and optimizing contingency management-based approaches for substance use treatment. CM-based approaches use financial reinforcement to increase the likelihood of clinical response [28]. CM methods are both effective and underutilized in substance use management. We believe that if dual users find it harder to quit yet the benefit to everyone is greater when they quit, it may be reasonable to compensate them extra when they successfully quit. In effect, the tandem use of both cg05575921 and ATS may give CM-oriented clinicians a rational basis through which to tune the effectiveness of their reward paradigm and optimize the likelihood of treatment success.

Given the fact that the reversion curve of cg05575921 methylation in response to smoking cessation is becoming increasingly refined, a natural question is “*what is the reversion curve of the ATS?*”. The answer is simply that we do not know. Methodological challenges are at least partially to blame for this lack of understanding. Defining the dose response of cg05575921 methylation to daily cigarette consumption and cessation greatly benefitted from the availability of two existing, readily employable, short-term biomarkers of smoking—exhaled carbon monoxide and cotinine—to help define the set point for non-smokers and to invalidate data from subjects who falsely stated that they achieved smoking cessation [20]. This was not a trivial problem. While collecting the four cohorts of subjects included in a recently published meta-analysis of the dose responsiveness of cg05575921, we found that ~4% of the subjects who enrolled and stated that they were lifetime non-users of nicotine-containing products in fact had significant serum levels of cotinine [20]. This high rate of unreliable self-report is a problem for the examination of alcohol consumption. We really do not have other highly sensitive biomarkers, which could be developed upon. The two most commonly used biomarkers for alcohol consumption, PEth and CDT, are also indicators of long-term use and do not have the sensitivity to robustly detect episodic relapses. Ethyl glucuronide (EtG) has a more favorable window of detection for this type of clinical study [14]. However, it is also poorly sensitive to lower levels of alcohol consumption. Because we serially assessed subjects undergoing inpatient alcohol treatment for our initial studies, and it is unlikely that those subjects were able to drink as inpatients, we obtained information about the reversion of the alcohol-induced signature over the first 3 to 4 weeks after stopping heavy drinking. However, those inpatient subjects represent the extreme spectrum of alcohol consumption, and findings from their methylation reversion may not generalize to lower ATS values. If defining the reversion curve for ATS values is essential for clinical or research purposes, it may be necessary to use newly developed, alcohol-sensitive “wearables” to verify the abstinence of subjects participating in those efforts.

In summary, in this preliminary study of a treatment population, we report that the smoking cessation is associated with marked decreases in an objective measure of alcohol consumption, even in the absence of therapies tailored to comorbid alcohol use. Limitations of the findings include a lack of self-report data on the clinical characteristics of the subjects’ alcohol consumption. We suggest that future efforts use objective alcohol consumption information to tailor smoking cessation therapy that could lead to improved rates of cessation for both forms of addiction.

## Figures and Tables

**Figure 1 genes-13-00002-f001:**
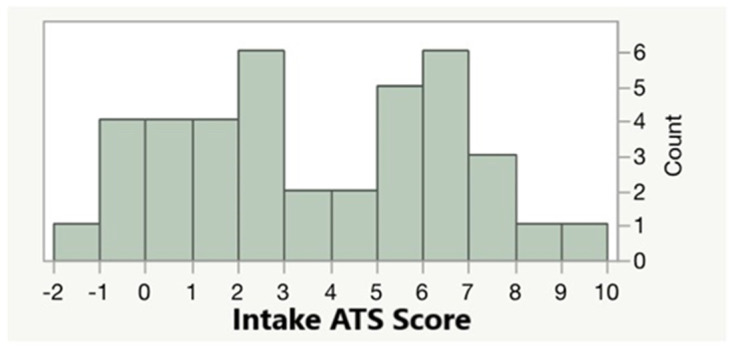
The distribution of ATS values at study intake. The distribution of the ATS in abstinent individuals is centered on zero, and scores above 4 are highly predictive of heavy alcohol consumption.

**Figure 2 genes-13-00002-f002:**
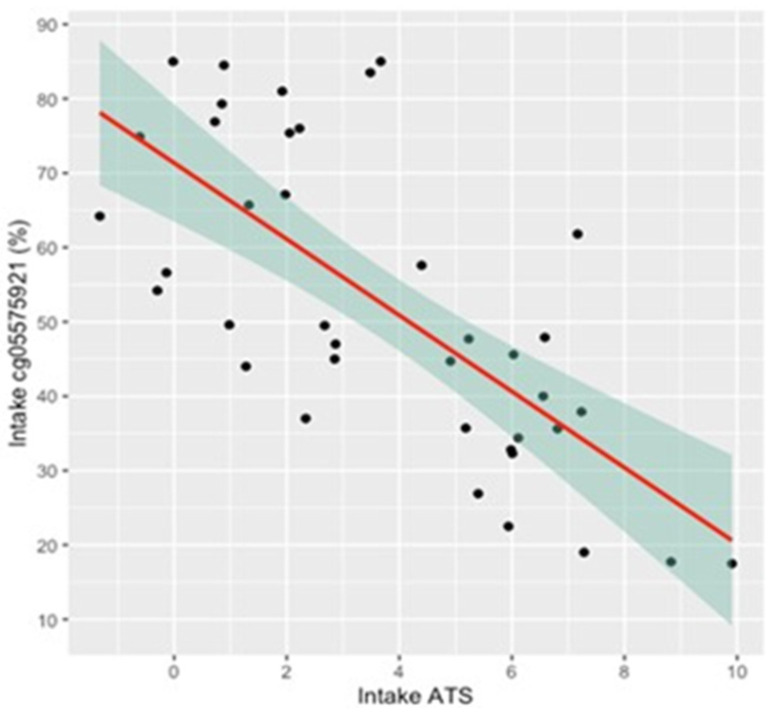
The relationship between intake cg05575921 methylation and ATS score. Intake cg05575921 is highly correlated with intake ATS (*n* = 39, *p* < 0.0001; adjusted *R*^2^ = 0.49). Shaded area represents the 95th percentile confidence zone.

**Figure 3 genes-13-00002-f003:**
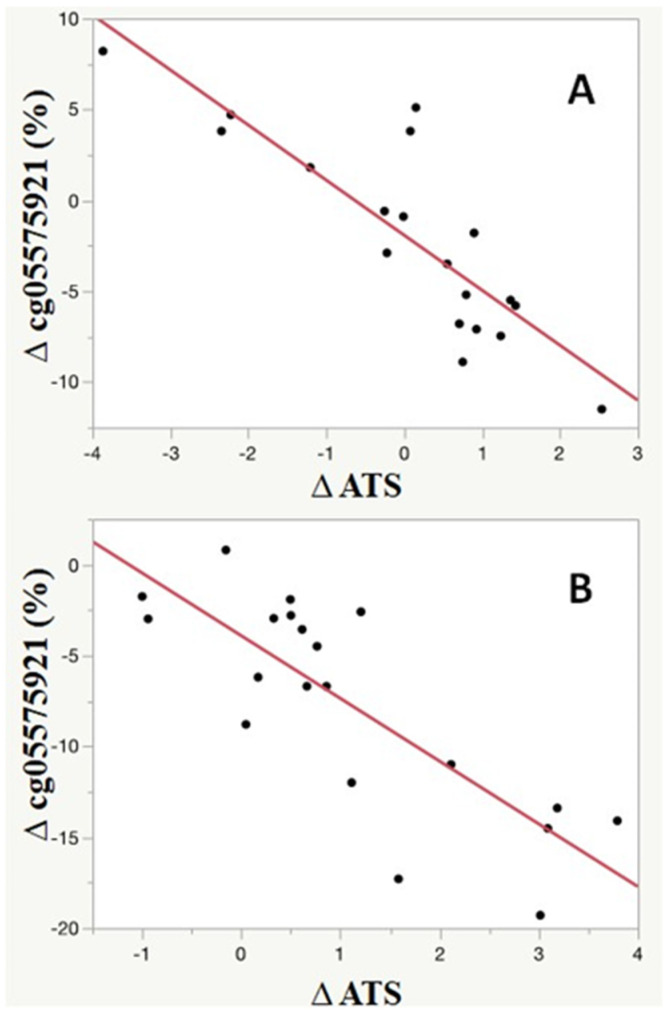
The relationship between the 90-day changes in cg05575921 methylation and ATS values for those who did not fully quit smoking (**A**; *n* = 19) and those who did fully quit smoking (**B**; *n* = 20). Although the overall magnitude of remethylation was greater in the quitters (**B**), the change (∆) in ATS score as a function of reduction in Scheme 05575921 was highly correlated with ATS values for both quitters (**B**, *n* = 20, *p* < 0.0001; adjusted *R*^2^ = 0.62) and non-quitters (**A**, *n* = 19, *p* < 0.0001; adjusted *R*^2^ = 0.70).

**Table 1 genes-13-00002-t001:** Demographic and clinical characteristics of the subjects.

	Non-Quitters	Quitters
**N**	19	20
**Age (years)**	45 ± 10.2	39.8 ± 9.9
**Gender**		
Male	11	11
Female	8	9
**Ethnicity**		
White	17	20
African American	1	
Other	1	
**Smoking Variables**		
Pack year consumption	34 ± 25	22 ± 10
Cigarettes per day	19 ± 13	16 ± 6
**cg05575921 Methylation (%)**		
Intake	47.2 ± 18.3	57.1 ± 22.1
Exit	49.4 ± 17.7	64.7 ± 17.9
Cotinine (ng/mL)	278 ± 135	206 ± 93
**Alcohol T Scores (ATS)**		
Intake	4.22 ± 2.95	3.26 ± 2.78
Exit	4.15 ± 3.03 *	2.18 ± 2.33

* different from quitter *p* < 0.05.

## Data Availability

The datasets used during the current study are available from the corresponding author on reasonable request.

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
