# Peer review of "Alcohol Use Intensity Decreases in Response to Successful Smoking Cessation Therapy"

_genes, 2021, doi:10.3390/genes13010002_

Round 1

Reviewer 1 Report

Dear Authors.

In my opinion this article is sufficiently novel and interesting. Presented article brings new, meaningful information. Nowadays addiction has become the more and more intensive and problematic for all over the word. New inside into this problem which was shown in the manuscript is good, especially when traditional addiction assessment tools are combined with the epigenetics.

I only would like to point a slight  mistakes in text: 

Page 6, line 201 and page 7 line 268  there is “in” used twice.

Author Response

Comment:in x 2” in line 202.

Response:  We have corrected this oversight on our part.

Reviewer 2 Report

Dear Authors,

The study is important and very interesting in terms of patients status health and therapy. In this study, it can be seen the potential of smoking therapy and alcohol dependence treatment .

Although, this study is interesting, the authors could be to change of several a elements:

Please try to avoid using the word "alcoholism". That word is becoming more stigmatized - lines 28 and 220.

Authors can add to the topic phrase "preliminary study" because this research needs a continuation.

At the beginning of manuscript the authors should highlight of the study aim.

In the methods - if the authors have data about a interview with participants, such as used scales, for example SADD for assess intensity of dependence. They can show this psychological scales  (possible that the scales score are conected for example with outcomes of ATS).
In the discussion - my recommend is description of a study limitation.

In the discussion - the line 205, the authors showed the results of another research. In this place authors should put a citation or write that this is own research (a data not published).

Author Response

Comment: “Please try to avoid using the word "alcoholism". That word is becoming more stigmatized - lines 28 and 220”.

Response:  The Reviewer makes a good point.  We have changed the wording in those lines to eliminate the term.   The term “alcoholism” is now only found in the reference section in the title of journals and articles.

Comment:Authors can add to the topic phrase "preliminary study" because this research needs a continuation.”.

Response:  We do not understand the term “topic phrase.”  Nonetheless, we now note in Line 268 that this is a preliminary study.

Comment:At the beginning of manuscript the authors should highlight of the study aim.”

Response:  At the conclusion of the introduction in line 78, we now state “The aim of this study is to test whether these prior findings are generalizable to other groups, such as clinical populations.”

Comment: “if the authors have data about a interview with participants, such as used scales, for example SADD for assess intensity of dependence....”

Response:  The Reviewer makes a good point.  We only have the data that we report. But this limited data is both a strength and a limitation.  In many ways, because we did not interrogate the subjects with respect to their alcohol use, it is unlikely that the alcohol patterns would change as a function of a Hawthorne effect.   However, it also makes understanding the context and impact of that use  difficult.  We have added the statement “Limitations of the findings include a lack of self-report data on the clinical characteristics of the subjects’ alcohol consumption” at line 270 to comply with the Reviewer’s request.

Comment:In the discussion - the line 205, the authors showed the results of another research. In this place authors should put a citation or write that this is own research (a data not published)..

Response:  The Reviewer may be mistaken.  In line 205, we are speculating about the frequency of comorbid alcohol and smoking. We do not show the results of other research.